# Development of a Specific Nested PCR Assay for the Detection of 16SrI Group Phytoplasmas Associated with Sisal Purple Leafroll Disease in Sisal Plants and Mealybugs

**DOI:** 10.3390/plants11212817

**Published:** 2022-10-23

**Authors:** Guihua Wang, Weihuai Wu, Shibei Tan, Yanqiong Liang, Chunping He, Helong Chen, Xing Huang, Kexian Yi

**Affiliations:** 1College of Ecology and Environment, Hainan University, Haikou 570228, China; 2Environment and Plant Protection Institute, Chinese Academy of Tropical Agricultural Sciences, Haikou 571101, China; 3College of Forestry, Hainan University, Haikou 570228, China; 4Key Laboratory of Integrated Pest Management on Tropical Crops, Ministry of Agriculture and Rural Affairs, Haikou 571101, China; 5Sanya Research Institute, Chinese Academy of Tropical Agricultural Sciences, Sanya 572025, China; 6Hainan Key Laboratory for Monitoring and Control of Tropical Agricultural Pests, Haikou 571101, China

**Keywords:** sisal purple leafroll disease (SPLD), *Dysmicoccus neobrevipes*, phytoplasma, specific primers, 16Sr RNA gene

## Abstract

Sisal purple leafroll disease (SPLD) is currently the most destructive disease affecting sisal in China, yet its aetiology remains unclear. In our previous research, it was verified to be associated with phytoplasmas, and nested PCR based on the 16S rRNA gene using universal primers R16mF2/R16mR1 followed by R16F2n/R16R2 was confirmed as the most effective molecular method for the detection of phytoplasmas associated with SPLD (SPLDaP). However, the method has a shortcoming of inaccuracy, for it could produce false positive results. To further manage the disease, accurate detection is needed. In this study, we developed a specific nested PCR assay using universal primers R16F2n/R16R2, followed by a set of primers designed on 16Sr gene sequences amplified from SPLDaP, nontarget bacteria from sisal plants, and other phytoplasma subgroups or groups. This established method is accurate, specific, and effective for detection of 16SrI group phytoplasma in sisal, and its sensitivity is up to 10 fg/μL of total DNA. It also minimized the false positive problem of nested PCR using universal primers R16mF2/R16mR1 followed by R16F2n/R16R2. This method was further used to verify the presence of phytoplasma in *Dysmicoccus*
*neobrevipes*, and the results showed that *D. neobrevipes* could be infected by SPLDaP and thus could be a candidate for vector transmission assays.

## 1. Introduction

Sisal (*Agave* spp.) is the most important hard fibre crop in the world [1]. Unfortunately, worldwide, it is infected by numerous fungal pathogens, including *Aspergillus welwitschiae* [1], *Phytophthora nicotianae* [2], and *Neoscytalidium dimidiatum* [3], and it is infested by insects such as *Dysmicoccus neobrevipes* [4] and *Scyphophorus acupunctatus* [5]. Purple leafroll disease (SPLD) is currently the most destructive disease affecting sisal in China. The symptoms of SPLD include a purple colour and rolling appearing from the leaf apex to leaf margin, followed by the appearance of yellow spots and withering of the entire leaf. This disease was first discovered in Hainan province of China in 2001, and since then, it has spread widely on several main sisal farms in Hainan and Guangdong provinces, causing a greater than 30% loss in sisal production per year (Kexian Yi, Environment and Plant Protection Institute, Chinese Academy of Tropical Agricultural Sciences, Haikou, China). Recently, this disease broke out in Pubei County within Qinzhou City of Guangxi province, representing a serious threat to the main sisal-producing region in China. However, the aetiology of this disease remains unclear.

From symptom analysis, it was found that SPLD had similar symptoms to some other plant diseases caused by viruses [6,7,8], phytoplasma [9,10,11], or nutrient deficiency [12,13,14]. Over the past several years, our attempts to study the aetiology of SPLD focusing on viruses and nutrient deficiency have yielded only minor results. Interestingly though, phytoplasma was verified to be present and associated with SPLD in sisal in our previous research (unpublished), and it is a member of ‘*Candidatus* Phytoplasma asteris’ (16SrI-B subgroup) [15]. Nevertheless, a causal link between SPLD and phytoplasma is still not established via Koch’s postulates. Therefore, effective prevention and control strategies still cannot be formulated and implemented.

An accurate method for the detection of phytoplasmas in host plants and vectors is important; it would play a key role in disease diagnosis. So far, various molecular methods for phytoplasma detection have been developed [16]. In particular, nested PCR assays using universal primers (e.g., P1/P7 or R16mF2/R16mR1, followed by R16F2n/R16R2) based on the 16S rRNA gene have been widely used for phytoplasma detection in different host plants [17]. This assay involves two rounds of amplifications via PCR using universal or group specific primers and has higher sensitivity than direct PCR [16]. However, it sometimes amplifies nontarget bacterial species in some host plants and vectors, due to the high conservation of the 16Sr RNA gene among various species of bacteria, causing a problem of false positives [18,19,20,21]. To avoid misdiagnosis, an accurate and effective PCR system for phytoplasma detection needs to be established.

In our previous research, nested PCR using universal primers R16mF2/R16mR1 and R16F2n/R16R2 was identified as the most effective molecular method for detecting phytoplasma associated with SPLD (SPLDaP) in sisal, but it produced many false positive amplicons of ~1.2 kb from nontarget bacterial species that had a similar size to the phytoplasma sequences generated via PCR assay (unpublished). Therefore, in this study, we attempted to (i) collect the SPLDaP sequences and false positive sequences of ~1.2 kb from sisal plants from the main sisal-growing regions of China, (ii) design several sets of nested PCR primers based on the F2nR2 regions of collected sequences, and (iii) select one set of specific primers to establish an accurate and effective nested PCR method for SPLDaP detection in sisal. Then, this method was used to identify the potential vectors of SPLDaP.

## 2. Results

### 2.1. Design and Screening of Specific Primers

All the samples collected from 80 SPLD-affected plants and 95 asymptomatic plants were amplified via nested PCR assay using the universal primers R16mF2/R16mR1, followed by R16F2n/R16R2. The results showed that there was a significant multiple nonspecific amplification, accounting for 93% of total samples; these nonspecific amplifications were observed in both SPLD-affected plants and healthy plants (Figure 1A-I). A total of 70 DNA bands of ~1.2 kb, based on constant and sharp DNA bands in 1% agarose gel, were cut from the gels and selected for sequencing, including 30 from symptomatic samples and 40 from asymptomatic samples. Sequence analysis indicated that 25 (21 from symptomatic samples and 4 from asymptomatic samples) of the 70 bands were phytoplasma sequences sharing 99.68–99.92% similarity, verified to be 16SrI-B via visual RFLP analyses using *i*PhyClassifier, and one of them was selected and deposited in GenBank under accession no. MW364375. The remaining 45 bands were false positive sequences (9 from symptomatic samples and 36 from asymptomatic samples), and they were divided into 11 different sequences from different species of nontarget bacteria (amplicon size: 1229 bp–1338 bp), which shared 90.51% to 99.45% similarities with multiple *Bacillus* strains (accession nos. KC494326.1, CP021434.1, KT720071.1, KM108715.1, JX505040.1), *Veillonella* strain (accession no. KP944173.1), and other uncultured bacterial 16S ribosomal RNA genes (accession nos. LC349042.1, LR641155.1, JQ028130.1, AB696041.1).

The F2nR2 regions of the collected phytoplasma and non-phytoplasma sequences were aligned, and four regions were selected for primer design (Appendix A). Based on the sequence alignment of the primer regions, there was 100% identity among all strains (ASS1, SPLDaP1 to SPLDaP9) of the 16SrI-B subgroup phytoplasmas recorded from diseased sisal plants of the target regions. In addition, the primer regions shared a high level of identity with other 16SrI subgroups and had a big difference with the 16Sr group outside the 16SrI group and the false positive sequences amplified from sisal plants. Two forward primers were ASF1 (5′-CAATAGGTATGCTTAGGGAGGAG-3′) and ASF2 (5′-TAGGGAAGAATAAATGATGGAAAA-3′), and two reverse primers were ASR1 (5′-CACTGGTTTTACCCAACGTTTA-3′) and ASR2 (5′-GCAACTGATAACCTCCACTG -TGT-3′). The corresponding product sizes were 632 bp (ASF1/ASR1), 817 bp (ASF1/ASR2), 399 bp (ASF2/ASR1), and 584 bp (ASF2/ASR2).

Nested PCR using the universal primers R16mF2/R16mR2 or R16F2n/R16R2, followed by the four candidate primers (ASF1/ASR1, ASF1/ASR2, ASF2/ASR1, and ASF2/ASR2), were separately used to detect phytoplasmas from the symptomatic and asymptomatic samples. The amplicons are shown in Figure 2. There was a single product of predicted size amplified by the four nested PCR primer sets. Sequenced results showed that all the amplifications were phytoplasma sequences that shared 100% sequence identity with multiple phytoplasma strains (e.g., accession nos. OP143676.1, ON325390.1, ON756116.1). Finally, primer sets R16mF2/R16mR2 or R16F2n/R16R2 followed by ASF1/ASR1 showed the best amplification efficiency. Then, nested PCRs using primers R16mF2/R16mR2 followed by ASF1/ASR1 and R16F2n/R16R2 followed by ASF1/ASR2 were used to test more samples. The results (Figure 3) revealed that amplicons using primers R16F2n/R16R2 followed by ASF1/ASR2 had higher amplification ratios, indicating that they are more suitable for SPLDaP detection.

### 2.2. Optimization of the PCR Detection Method and Analysis of Assay Sensitivity

A series of annealing temperatures (49–64 °C) of candidate-specific primers ASF1/ASR1 were tested. The results showed that the annealing temperature had little effect on the amplification efficiency (Figure 4A). To ensure the specificity and efficiency of the amplification products, 58 °C was selected as the annealing temperature for subsequent PCR amplification. The final PCR program of nested specific primers ASF1/ASR1 was as follows: 94 °C for 3 min; 35 cycles at 94 °C for 30 s, 58 °C for 30 s, and 72 °C for 1 min; followed by 72 °C for 10 min.

To analyse the sensitivity of the optimized nested PCR, a 10-fold dilution series (10^0^ to 10^−6^) of the total DNA of SPLDaP-infected sisal plant samples with an initial concentration of 1 ng/μL was used as templates. The results (Figure 4B) revealed that there was no amplification after the 10^−6^ dilution, which means the method can detect at least 10 fg/μL of total DNA of SPLDaP-infected sisal samples.

### 2.3. Evaluation of Specificities

Both the new method and conventional nested PCR using universal primers R16mF2/R16mR1 followed by R16F2n/R16R2 were separately used to test more samples from different sampling sites. Representative results are shown in Figure 1A. Compared with conventional nested PCR (Figure 1A-I), there was a single product of predicted size and no false positive amplifications for sisal samples from different sampling sites using the new method (Figure 1A-II). Therefore, this new method was effective and geography-specific for phytoplasma detection in sisal.

Further, the two methods were separately used to test different tissues, including leaves, stems, and roots, from the SPLD-affected sisal plants and healthy sisal plants. The results are shown in Figure 1B. Compared with conventional nested PCR (Figure 1B-I), there was a single product of predicted size and no false positive amplifications using the new method (Figure 1B-II). Therefore, this new method was also tissue-specific for phytoplasma detection in sisal.

### 2.4. SPLDaP Detection in Mealybugs

The field survey showed that the potential vectors of phytoplasmas were seldom observed in sisal fields. Interestingly though, mealybugs (Figure 5A) were found to be widespread in the sisal-planting regions affected by SPLD, seriously infesting sisal. To determine the taxon of the mealybugs, specific DNA markers were used in PCR, as previously described [22]. The phylogenetic analysis of the obtained 28S gene sequence fragment revealed that the mealybug was clustered with *Dysmicoccus neobrevipes* (Appendix A), with 99% nucleotide identity (Appendix A). The morphological characteristics of the mealybug appeared to be similar to those of *D. neobrevipes*.

To verify the presence of phytoplasmas in *D. Neobrevipes*, the new method was used to detect phytoplasmas in eleven mealybug samples and the corresponding plant leaf samples from three different sites. The results showed that phytoplasmas were detected in 7 of 9 sisal plant samples and 5 of 9 mealybug samples (Figure 5B). Further sequencing indicated that the amplified DNA fragments from the mealybugs were phytoplasma-specific and shared more than 98% identities with the corresponding sequences obtained from the plants. These results showed that the new method was equally suitable for phytoplasma detection in *D. neobrevipes* from sisal plants, and phytoplasma was verified to be present in *D. Neobrevipes*.

Then, the mealybug and sisal samples that were positive for phytoplasmas were amplified via nested PCR using primers R16mF2/R16mR2 and R16F2n/R16R2. Sequence analysis revealed the 16S rDNA sequences (F2nR2 regions) of phytoplasma from sisal plants and mealybugs shared 99.60–99.92% similarity with each other and 98.88–100% similarity with the SPLDaP recorded from diseased sisal plants of the F2nR2 region (MW364375-ON921305). Sequences analysis, phylogenetic analysis, and virtual RFLP analyses using *i*PhyClassifier further indicated that the phytoplasmas from the mealy- bug and sisal samples equally belonged to 16SrI-B (Appendix A). Therefore, this suggested that *D. neobrevipes* can ingest SPLDaP, and this species could be a candidate for vector transmission assays for SPLDaP.

## 3. Discussion

SPLD is a destructive disease in China, and it has caused a significant economy loss for sisal production. The presence of phytoplasma and its association with SPLD suggested that phytoplasma was a possible pathogen of SPLD. However, the inaccuracy of the nested PCR method using universal primers R16mF2/R16mR1 and R16F2n/R16R2 for detection of SPLDaP suggested the need to develop a specific method for further disease diagnosis. To date, an array of molecular methods has been developed for the detection of phytoplasmas, including PCR amplification, nested PCR, closed tube quantitative PCR assays, ddPCR, and the loop-mediated isothermal amplification (LAMP) assay [16], and more non-ribosomal genes, such as *RpoB* [23], *SecY* [24], *Tuf* [25,26], and *Cpn60* [27] have been used as targets. However, the time consumption, cost, labour, laboratory equipment requirement, personal experience, and detection accuracies and efficiencies varied for these published methods [16]. In all, nested PCR based on the 16S rRNA gene is still the most basic molecular detection method of phytoplasma [16], especially for the diagnosis of new phytoplasma disease, although it sometimes amplifies nontarget bacterial species due to the high conservation of the 16Sr RNA gene among various bacteria [18,19,20,21]. In this study, we collected sequences of the 16Sr RNA gene from these nontarget species as well as those of SPLDaP, 18 subgroups of the 16SrI taxonomic group, and 30 other 16Sr groups from GenBank (Appendix A); based on these sequences, an accurate and effective nested PCR method was developed and evaluated. This method could detect phytoplasma-infected sisal samples with a minimum concentration of 10 pg/mL, which is sufficient for the detection of sisal plant samples.

Sampling is also important for the diagnosis of phytoplasma. Since the phloem is the niche of phytoplasmas, plant organs with abundant vasculature, such as the flowers, leaves, stems, and roots, are likely areas of phytoplasma presence [28]. However, numerous reports showed that phytoplasmas are unevenly distributed in the host plants they infect; the patterns are phytoplasma-host specific and vary from season to season [29,30,31,32,33]. Therefore, the appropriate plant parts can be sampled to increase the detection efficacy for developing detection methods in the field for research purposes. For example, sampling from multiple tissues is more suitable in the screening of phytoplasma-free plants and evaluation of transmission experiment results because this may avoid false negatives caused by sampling from a single tissue; sampling from the plant parts with a high phytoplasma ratio in host plants will aid transmission studies to focus on vector discovery and ecology. In this study, the new method was not only suitable in the mixed samples, including leaves, stems, and roots, but also in the single sample from a plant tissue; moreover, it was also specific for the detection of 16SrI-B group phytoplasmas. Based on primer design via nucleotide sequences (Appendix A), it should specifically detect other 16SrI subgroup phytoplasma in sisal plants in the future. Subsequently, this method could be used for screening of phytoplasma-free plants, vector discovery, and evaluation of transmission experiment results. Thus, it will be helpful for following the causal agent between SPLD and 16SrI-B group phytoplasmas in sisal.

The grey pineapple mealybug, *Dysmicoccus neobrevipes* Beardsley (Hemiptera: Pseudococcidae), is an important pest in *Agave* spp. [34]. Feeding by *D. neobrevipes* may cause leaf yellowing, defoliation, reduced plant growth and, in some cases, cause plant death [35]. It was also reported to be a vector for only a few viruses, such as PMWaV [36], grapevine leafroll viruses [37], and little cherry virus [38]. In China, feeding by *D. neobrevipes* was identified as the cause of SPLD in sisal because SPLD always occurs after mealybug infestation. In this study, phytoplasma was verified to be present in *D. neobrevipes*. Although the aetiology of SPLD is still unknown, the discovery of phytoplasma in sisal plants and *D. neobrevipes* will provide important information for pathogen identification. Currently, none of the described mealybugs are known to be a phytoplasma vector [39]. To further clarify whether *D. neobrevipes* is the vector of SPLDaP, more evidence will be required including an insect transmission assay and phytoplasma observation of the salivary glands of mealybugs via transmission electron microscopy. Nonetheless, this is the first reported instance of SPLDaP detection in *D. neobrevipes*.

To summarize, a new nested PCR method was established in this study, and it could specifically detect 16SrI-B group phytoplasma in sisal samples and *D. neobrevipes*. Thus, it could be used for following the causal agent between SPLD and 16SrI group phytoplasmas in sisal. However, it is still possible to produce a non-specific amplifications duo for limited-number samples for the collection of non-target sequences (~1.2 kb). In the future, the reservoir of non-target sequences of ~1.2 kb could be supplemented when new non-target sequences appear again, and more specific primers should be designed to meet the research purpose. Best of all, this is the first reported specific nested PCR for SPLDaP detection in sisal plants and *D. neobrevipes.* This method can provide reference for the development of other phytoplasma-specific detection methods in the future.

## 4. Materials and Methods

### 4.1. Plant Samples and Sequence Collection

Samples used for sequence collection were obtained from three provinces of Hainan, Guangdong, and Guangxi between 2018 and 2021. The location of sample collection sites is shown in Figure 6. A total of 80 symptomatic samples from the SPLD-affected sisal plants and 95 symptomatic samples from healthy sisal plants were collected (Appendix A). Leaf (0.1 g each), stem (0.1 g each), and root (0.1 g each) tissues from one sisal plant were mixed and were regarded as one plant sample (Figure 7). DNA was extracted from the sisal samples using a DNeasy Plant Mini Kit (Qiagen GmbH, Hilden, Germany), according to the manufacturer’s instructions.

All DNA was amplified via nested PCR assay using the phytoplasma universal primers R16mF2/R16mR1, followed by R16F2n/R16R2. The first PCR amplification was carried out in 20 μL reaction volumes containing 2 μL of 10× Ex Taq Buffer (Mg^2+^ Plus), 1.6 μL dNTP mixture (each 2.5 mM), 0.5 μL of each 10 μM primer, 0.2 μL Ex Taq (2 μL), 1 μL DNA, and 14.2 μL double-distilled H_2_O (sterile). For nested PCR amplification, the amplicons of the first PCR were diluted in a 1:5 ratio with double-distilled water (sterile) and 1 μL was used as a template. PCR reactions were performed in a Mastercycler (Eppendorf, Germany). The program, sequences for the primer pairs, and corresponding product sizes were based on previous studies [17]. Each PCR was repeated three times with all sisal samples to confirm and validate the amplification results. The phytoplasma detected previously in periwinkle (*Catharanthus roseus*) in our laboratory was used as a positive control [40]. Samples devoid of DNA templates were used as negative controls.

The final products amplified via PCR were subjected to agarose gel electrophoresis. The gels were stained with GoldView II and photographed under ultraviolet light. The target fragments of the amplified 16SrRNA gene from the nested PCR assay were cut from the gels using Multicolor Fluorescence/Chemiluminescence Imaging Analyzer (UVITEC, Cambridge, England) and purified using Wizard^®^ SV Gel and a PCR Clean-up System (Promega, Madison, USA). The purified DNA fragments were ligated to the pMD18-T simple vector (Takaka Bio Inc., Shiga, Japan) and sequenced bidirectionally (Sangon Biotech, Shanghai, China). Sequence alignment and manual revisions were achieved via BLASTn searching (National Center for Biotechnology Information; http://www.ncbi.nlm.nih.gov/) (accessed from 18 March 2018 to 31 December 2021) followed by comparison with phytoplasma sequences in GenBank. Identification of phytoplasmas was performed through sequence analysis, phylogenetic analysis using MEGA 6, and virtual RFLP analyses using *i*PhyClassifier [41]. All the sequences of ~1.2 kb, including phytoplasma and non-phytoplasma, were collected and used for primer designing.

### 4.2. Designing and Screening of Specific Primers

To obtain the specific primers for phytoplasma detection in sisal, the F2nR2 region of SPLDaP, all the subgroups of the 16SrI taxonomic group available on GenBank, and a subgroup from each of the existing 16Sr groups as well as the false positive sequences of ~1.2 kb amplified from sisal plants, were aligned using online software *Multalin* (http://multalin.toulouse.inra.fr/multalin/multalin.html) (accessed on 17 January 2022). This alignment process was to provide insight into the range of phytoplasmas that this assay could detect and to ensure that it would not amplify non-phytoplasma bacterial DNA (Appendix A). Specific primers were then designed based on the aligned sequences and analysed for the presence of secondary structures and the possibility of 3′ terminal pairings using the primer designing tool of Primer 5.

The PCR assays for screening of specific primers were performed using universal primer R16mF2/R16mR1 or R16F2n/R16R2 followed by candidate primers. The PCR reaction system and cycling conditions were the same as above, except the annealing temperature of the candidate primers, which was 61 °C. Specific primers were selected by comparing the specificity, amplification efficiency, and primer dimer of candidate primers.

### 4.3. Optimization of the Detection Method and Sensitivity Determination

To optimize the detection method, a series of annealing temperatures of the candidate primers were tested. Further, the analytical sensitivity of the assay was evaluated by testing 10-fold serial dilutions of SPLDaP-infected sisal DNA. The DNA sample in this study was known to be positive for phytoplasma 16SrI-B, verified via RFLP analyses using *i*PhyClassifier. The optimum annealing temperatures and the detection limit were determined using three replicates. The nested PCR product of the sample was purified and sequenced.

### 4.4. Evaluation of Primers Specificity

To evaluate the specificity of the new method, the new nested PCR and conventional nested PCR assays were compared using 16S rDNA specific primers and universal primers R16mF2/R16mR1 followed by R16F2n/R16R2, respectively. Eighty symptomatic and eighty asymptomatic samples collected from the different sites were used to evaluate the geographic specificity of the new method. More samples of leaves, stems, and roots were respectively collected from 12 asymptomatic and 21 symptomatic plants in different sampling sites and were used to evaluate the tissue specificity of the new method.

### 4.5. Identification of Potential Vectors

For the identification of potential vectors of SPLDaP, the known insect vectors of phytoplasmas, including psyllids, leafhoppers, and planthoppers, were searched for in the sisal fields. These insects were identified based on morphological and molecular characters. Total DNA was extracted from these insects using a TIANamp Genomic DNA Kit (TIANGEN Biotech, Beijing, China), according to the manufacturer’s instructions. The new method was used to verify the presence of phytoplasma in the insects. The nested PCR assays using universal primers R16mF2/R16mR2 and R16F2n/R16R2 were used to detect phytoplasma in potential vectors for phytoplasma identification; the methods of phytoplasma identification were the same as above.

## Figures and Tables

**Figure 1 plants-11-02817-f001:**
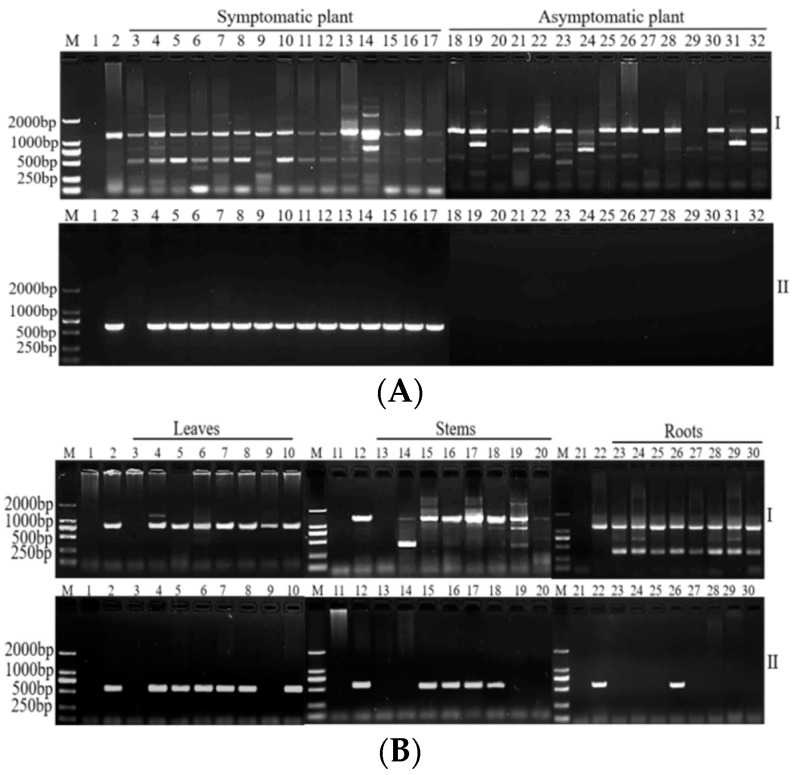
Results of mixed sisal samples (root, stem, and leaf) from different sampling sites (**A**) and single sisal samples (root, stem, or leaf) from plant tissue (**B**) detected via the new method (II) and conventional nested PCR (I). (**A**) Lane M: marker DL2000; lane 1: negative control; lane 2: positive control; lanes 3–17: symptomatic samples; lanes 18–32: asymptomatic samples from healthy plants. (**B**) Lane M: marker DL2000; lane 1: negative control; lane 2: positive control; lanes 3, 13, 23: the samples of leaf, stem, and root from healthy plants; lanes 4–10, 14–20, 24–30: the samples of leaf, stem, and root from symptomatic samples.

**Figure 2 plants-11-02817-f002:**
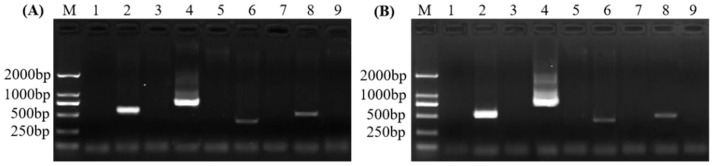
Results of nested PCR using the universal primer (**A**) R16mF2/R16mR2 or (**B**) R16F2n/R16R2, followed by four pairs of candidate-specific primers for SPLDaP detection in sisal. The even numbers (2, 4, 6, 8) of lanes are SPLDaP-positive samples, and the odd numbers (3, 6, 7, 9) of lanes are SPLDaP-negative samples. Lane M: marker DL2000; lane 1: negative control; lanes 2–3 (**A**,**B**): specific primers ASF1/ASR1; lanes 4–5 (**A**,**B**): specific primers ASF1/ASR2; lanes 6–7 (**A**,**B**): specific primers ASF2/ASR1; lanes 8–9 (**A**,**B**): specific primers ASF2/ASR2.

**Figure 3 plants-11-02817-f003:**
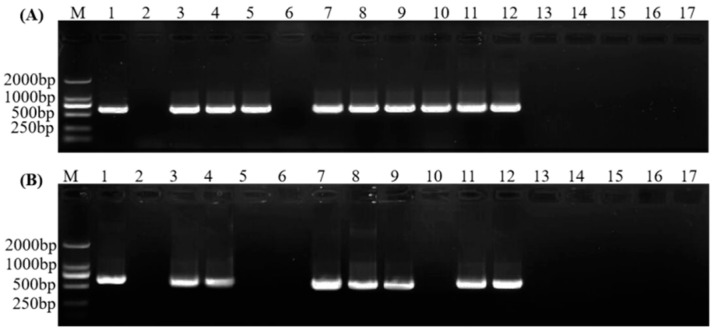
Results of nested PCR using primers (**A**) R16F2n/R16R2 followed by ASF1/ASR1 and (**B**) R16mF2/R16mR2 followed by ASF1/ASR1 for SPLDaP detection in sisal samples. Lane M: marker DL2000; lane 1: positive control; lane 2: negative control; lanes 3–12: symptomatic samples; lanes 13–17: asymptomatic samples.

**Figure 4 plants-11-02817-f004:**
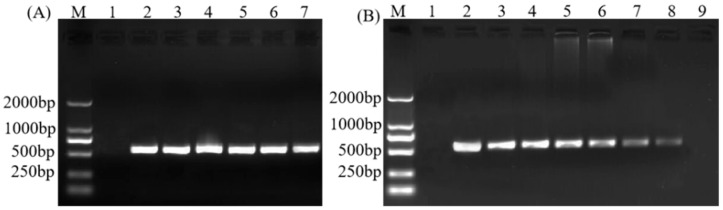
Results of SPLDaP-specific nested PCR using primers R16F2n/R16R2, followed by ASF1/ASR1 with (**A**) different annealing temperatures and (**B**) different template concentrations. (**A**) Lane M: marker DL2000; lane 1: negative control; lanes 2–7: annealing temperature 49 °C, 52 °C, 55 °C, 58 °C, 61 °C, 64 °C, respectively; (**B**) Lane M: marker DL2000; lane 1: negative control; lane 2: positive control; lanes 3–9: DNA dilution 10^0^–10^−6^ (1 ng/μL, 100 pg/μL, 10 pg/μL, 1 pg/μL, 100 fg/μL, 10 fg/μL, 1 fg/μL), respectively. Total DNA extraction from SPLDaP-infected sisal as a template for all reactions.

**Figure 5 plants-11-02817-f005:**
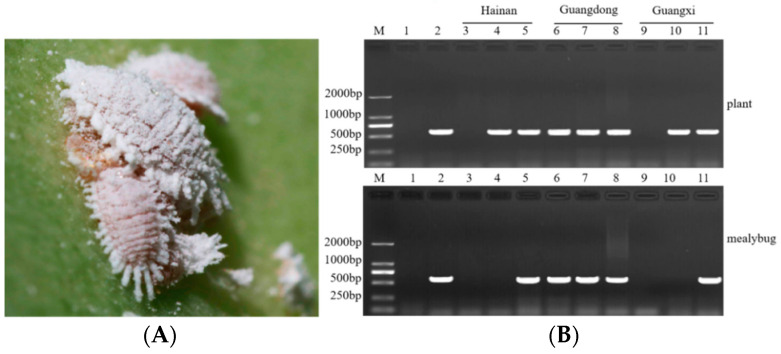
(**A**) Mealybugs infesting leaves of sisal. (**B**) Results of phytoplasma amplified from SPLD leaf samples and corresponding mealybugs samples using the new method. Samples were collected from different locations including Hainan province, Guangdong Province, and Guangxi Province. Lane M: marker DL2000; lane 1: negative control, lane 2: positive control; lanes 3–11: the samples of SPLD leaf and corresponding mealybugs collected from Hainan (lanes 3–5), Guangdong (lanes 6–8), and Guangxi (lanes 9–11) provinces.

**Figure 6 plants-11-02817-f006:**
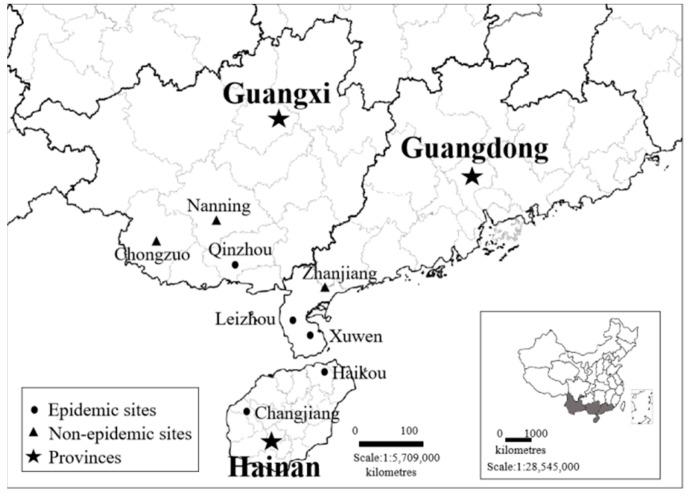
Location of sample collection sites in south China between 2018 and 2021.

**Figure 7 plants-11-02817-f007:**
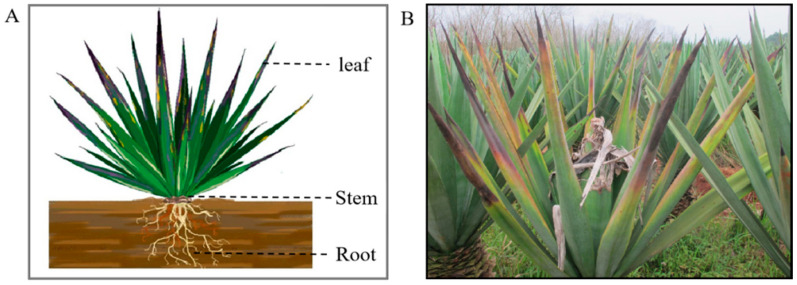
(**A**) Schematic diagram of the sampling parts on the sisal plants. Dotted lines indicate the sampling tissues; (**B**) sisal plants infected with PLD, showing purple margins, rolling, and yellowing of leaves accompanying central head rotting.

## Data Availability

DNA sequences are available in the GenBank database, with the accession numbers listed in the Results. All other relevant data are within the paper and Appendix A.

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
