# Peer review of "Development of a Specific Nested PCR Assay for the Detection of 16SrI Group Phytoplasmas Associated with Sisal Purple Leafroll Disease in Sisal Plants and Mealybugs"

_plants, 2022, doi:10.3390/plants11212817_

Round 1

Reviewer 1 Report

Article is well written, almost polished. Results mostly clearly presented, I only missed clarification/confirmation that you detected only I-B phytoplasma in your samples.

First report of this phytoplasma in mealybug is significant finding. Don’t be afraid to state it more loudly.

I have some problem only with the way you present the “neediness “of your work and how you try to “sell it”. In introduction you greatly over state the problem of universal PCR primer pairs.

You do not provide substantial evidence for that. These primers were used for decades now. You are not miracle workers suddenly coming to save the world from some big problem. The number of publications where those primers worked fine will dwarf those few that complained.
So present everything as it is. They didn’t work well for you, so you needed new ones for your situation. Whether anyone else will use your primers for detection of Sr-I group phytoplasmas in other plants remains to be seen. And seriously use better quality water for PCR in the future.

2-3: First detection of phytoplasma in mealybug (at least I understood it is first report of it) is bigger achievement than testing new primers in my book. And you need to advertise it more strongly. Maybe it is possible to modify article name to represent that.

39-42: Your article is about phytoplasmas, not viruses. You present the SPLD here and cite one article that is specifically about “purple leaf roll virus”. Yes, later you say that SPLD might be caused by several reasons. But reference [6] is in Chinese, I cannot read it, and its name clearly implies that SPLD was caused by a virus. So here you are talking about 30% yield loses caused by virus, and using that as justification for phytoplasma research… Either clarify much more what actually was discovered in reference [6] or do not use virus caused disease devastation as reason for phytoplasma investigation.

45-46: This sentence is nonsense in current way it is written. You talking about SPLD but you cite articles [7-9] about broad bean wilt virus, brassica yellows virus, potato leafroll virus and bacteria causing specific different diseases like PYLR. SPLD is not equal to PYLR. Your sentence now says “SPDL is caused by viruses and here is proof [7-9]”. Which is not true, those articles are not about SPDL. If you wanted to say that “several different things can cause similar morphologic effects/disease-like symptoms (purple color, leaf rolling) on plants” then you need to rephrase sentence completely.

46-48: You say “Over the past several years, studies of the aetiology of SPLD focusing on viruses and nutrient…”, plural. As if a lot of different research was made. But then you cite just one publication [6]. And that one is your local value, done in Chinese… So, you fail to give proof that many relevant and significant studies actually were made. Provide more proof or use much more modest expressions.

63: only reference [21] has mention of false positives with nested PCR. Publication [20] is not about it and is not fitting as reference in here.

61-63: Whole sentence is a bit “too strong”. False positives is not that of a big of a problem in general as you make it sound. Your phrasing sounds like “this is main problem for everyone in the world and so we going to save the world by fixing it”. For one, some level of non specific products will always be present in nested pcr. Secondly nested pcr is needed because of low dna concentrations so nobody is going to replace it at the moment (when cost effectiveness is considered). Thirdly, you created species specific primers for very limited applicability. So, the solving of false positives problem in nested PCR is to make separate primes specific to every phytoplasma subgroup? That is mildly saying very unpractical and not efficient.

You should rephrase it to represent the real truth more, which is that YOU had a problem with universal primers when dealing with one particular plant disease, so you needed to make more optimal detection method for your situation.

If you want to make bold claims then you need to give much more evidence, like at least 6 or 7 references. Out of your provided two, one [20] is not even fitting. But the fact remains, that for every single article where people had non specific amplicons with universal primers, you can have at least twenty other articles where those primers worked fine and all was ok…

Also, articles like this one are focused on the matter more and are better references: “Difficulties with conventional phytoplasma diagnostic using PCR/RFLP analyses”. (Not sure if I am allowed to suggest references, but I have no conflict of interests and are not involved with this one, or its authors)

63-64: Same as before. Rephrase to make it clear that you had a problem and needed solution for your specific plant of interest. Don’t make it sound that it is universal problem for everyone.

85: “verified to be 16SrI-B by RFLP analyses”. Don’t see why you need RFLP. You sequenced all of the 25 fragments. You can BLAST them. You know that they belong to Sr-IB subgroup by nucleotide sequence. There is no need for RFLP.

Figure 1. I would recommend moving figure1 to supplementary material. Majority of the readers will not be interested in sequences. Those few who will be can find it in supplementary material. Making such alignments is simple routine work and is unnecessary. We are interested in results of how your primers performed, not in the process of how you made those primers.

221-222: Again, RFLP is unnecessary. You had the whole nucleotide sequences. You knew the phytoplasma subgroub from nucleotide sequences. You see the sequence, you can compare it. You do not need to cleave it with restriction enzymes to further prove it.

237: you can add my suggested article (Difficulties with conventional phytoplasma diagnostic using PCR/RFLP analyses) as a reference next to [21]. Maybe find one more fitting publication to illustrate this issue better.

245-247: “…more methods can be developed in the future…”. I would delete this sentence. It sounds as anti-advertising of your method. You made good method and this sentence lessens your achievement.  Methods will always be a compromise of cost, time, sensitiveness and other factors, I doubt there can be one universally best way of doing particular task.

254: “seasona”

263: figure 3 does not have all kinds of different Sr-I subgroups. Based on the sequences that you used to create your primers, it could and should work. But in reality, you did not test them on all other subgroups. All your detected ptytoplasmas were Sr-IB, no? Maybe other subgroups don’t even infect sisal. So, you need to clarify it. That in practice you detected only IB.

You CANNOT claim that “…it was also specific for the detection of 16SrI group phytoplasmas”. You have no proof. You can say “we tested to detect IB, but based on design by nucleotide sequences it should detect all other group I subgroups”.

293-296: I don’t believe that you were getting false positives when using all of RpoB, SecY, Tuf and Cpn60 genes. Additional genetic markers like these are used to supplement 16Sr system for finer identification of similar strains of phytoplasmas. By design, extra markers like these are much more variable genes than ribosomal operon. Been working with many of them and can say that often it is problem to find universal primers fitting for whole phytoplasma genus, because sequences are very variable. I don’t see people having issue with non-specific amplicons of non phytoplasmal bacteria. Provide substantial proof that they are bad markers. In any case, mentioning that you were not successful with them raises many new questions. And your answer “believe us they bad (data not published)” does not convince anybody. So maybe better don’t even mention them if you have nothing substantial to say.

322, 323: “double-distilled” water? You don’t even say if its sterile. Simple double distilled water IS NOT sterile high-grade water (That could explain your non-specific products with genes like secY…). I don’t think that you are working in military grade sterility laboratory. Distilled water has plenty opportunity to collect dust and environmental bacteria from the air in your lab. I hope that you actually used for PCR and dilution a nuclease free water bought from your polymerase providers, and just used wrong name of it. If not, then please in the future use a properly sterile and clean water for PCR and primers. Buy that water from certified sellers. Nothing that is in laboratory, we have type1 mili-Q system, and even it is not good enough for PCR applications (especially nested PCR which sensitivity is so much higher). While you store it, handle it, transport it, pour into smaller containers, everywhere there are opportunities for it get contamination.

365-370: for the record, using your primers to test on fungi for non-specific amplification is a “bit pointless”… They are what used to be called eukaryotes, they don’t even have 16S ribosome. It would be interesting achievement if primers designed for bacterial 16S would work on eucaryotic 18S. And if you think their sequences are similar then why you did not include these fungi sequences in figure1 while you designed the primers.

Reviewer 2 Report

Dear Authors,

 The manuscript Development of a Specific Nested PCR Assay for the Detection of 16SrI Group Phytoplasmas Associated with Sisal Purple Leafroll Disease“ submitted as Article to Plants journal by Guihua Wang et al. The manuscript describes Sisal purple leafroll disease (SPLD) of Agave spp. and the presence of phytoplasma in Dysmicoccus Neobrevipes (grey pineapple mealybug) insects.

Specific for SPLDaP phytoplasma detection PCR primers and assay were developed. This established method is accurate, specific, and effective for detection of this pathogen in plants and insects. This is the first reported instance of SPLDaP detection in D.neobrevipes. None of the described mealybugs is known to be a phytoplasma vector and future research are needed for this case.

New data were obtained and new methodology for specific detection was established and described.

Manuscript is written very well. These important results could be published in Plants after very minor revisions.

 Line 327. Catharanthus roseus –  font italic

Line 482. Reference 23. Deividas,V.; Rasa, J.; Robert, E.D.  – Names of authors are wrong. They must be Valiunas D., Jomantiene R., Davis R.E.

 Sincerely, 2022 Sept 30

Reviewer 3 Report

My comments can be found in the attached MS.

Reviewer 4 Report

This is an excellent paper and I enjoyed reading it. Please ensure to make all figures are self explanatory. Make sure that all scientific names are all written in italic format. The references  section needs extra care and more revision please. Well Done and the paper can be accepted with minor revision. 
